# Bridging the Gap in Technology Transfer for Advanced Process Control with Industrial Applications

**DOI:** 10.3390/s22114149

**Published:** 2022-05-30

**Authors:** Vitali Vansovits, Eduard Petlenkov, Aleksei Tepljakov, Kristina Vassiljeva, Juri Belikov

**Affiliations:** 1Department of Computer Systems, Tallinn University of Technology, 12618 Tallinn, Estonia; vitali.vansovits@taltech.ee (V.V.); eduard.petlenkov@taltech.ee (E.P.); aleksei.tepljakov@taltech.ee (A.T.); kristina.vassiljeva@taltech.ee (K.V.); 2Department of Software Science, Tallinn University of Technology, 12618 Tallinn, Estonia

**Keywords:** advanced control, model predictive control, industrial process, Industry 4.0, technology transfer

## Abstract

In the present paper, a software framework comprising the implementation of Model Predictive Control—a popular industrial control method—is presented. The framework is versatile and can be run on a variety of target systems including programmable logic controllers and distributed control system implementations. However, the main attractive property of the framework stems from the goal of achieving smooth technology transfer from the academic setting to real industrial applications. Technology transfer is, in general, difficult to achieve, because of the apparent disconnect between academic studies and actual industry. The proposed software framework aims at bridging this gap for model predictive control—a powerful control technique which can result in substantial performance improvement of industrial control loops, thus adhering to modern trends for reducing energy waste and fulfilling sustainable development goals. In the paper, the proposed solution is motivated and described, and experimental evidence of its successful deployment is provided using a real industrial plant.

## 1. Introduction

Control systems are ubiquitous in industry [1]. At the same time, the way that these control systems are designed, configured, and deployed has been frequently criticized [2]. Indeed, it is commonplace that the deployed controllers underperform significantly in their intended application domain, which leads to considerable resource and energy waste, but can be improved through proper implementation and tuning using optimization [3].

Optimization is, in general, one of the key development trends in many areas including manufacturing and related industries. In part, this is motivated by the vision of the fourth industrial revolution—commonly referred to as Industry 4.0 [4]. From the perspective of optimization, major concerns are related to the efficiency of production processes, consumption of raw materials, utilization of assets, various emissions, etc. Meanwhile, it is not possible to do a full upgrade of existing industrial assets at once; consequently, to achieve the transition to sustainable and efficient modes of industrial operation, the analysis of utilized technologies, development of optimization solutions, and their deployment to existing manufacturing systems must take place. In fact, a vast amount of optimization algorithms has been developed throughout the years to address the mentioned issues at all levels of production processes [5]. New materials, mechanical solutions, and control algorithms can be implemented to mitigate the bottlenecks of a certain process to make production use fewer materials, produce less emissions, and make products of the highest possible quality.

As far as control systems are concerned, the majority of existing systems still use proportional-integral-derivative (PID) controllers. They are easy to set up and deploy, but they may also be difficult to tune, considering the number of control loops involved in a given production process. Indeed, this is a serious bottleneck that has its roots in general lack of knowledge and technology transfer (KTT) between academia dealing with control system research and the industry which heavily relies on such knowledge to improve all aspects of control system performance [2,3].

There are certain successful trends to improve existing PID loops performance utilizing Industry 4.0 techniques such as big data collection and cloud computing to collect control loop diagnostic data and analyze their performance [6]. On the other hand, other much more efficient control system methods are also readily available based on the *advanced process control* (APC) approach.

It is commonly acknowledged that the application APC in manufacturing can be profitable [7]. In recent years, there have been a number of developments in the area of APC to further confirm this. In [8], a real-time optimal control scheme for a chemical process is developed leading to significant improvements of production scheduling. The scheme is verified with real industrial data. On the other hand, in [9], the state-of-the-art of model predictive control is given. One of the claims in the conclusions states that one of the key issues to be addressed is the gap between the theory and practice of model predictive control. In [10], a comprehensive state-of-the-art of model predictive control in the application to electric machines and systems is provided. The authors again highlight the importance of MPC in industrial applications and predict that it will be more frequently applied to industrial problems in the future due to its favorable qualities. Additionally, in [11], a complete platform supporting the APC approach is presented, tailored to the requirements of Industry 4.0.

Similarly, in this paper, APC is considered as a tool that has the capacity to provide significant improvements in the context of industrial control and which would allow to achieve a smooth technology transfer from theory to practice, i.e., from academia to industry. Specifically, we address the problem of the practical implementation of advanced process control in the industrial space which is still largely dominated by PID control loops. Advanced process control techniques are considered to play a crucial role in achieving the goals of Industry 4.0 [12]. Naturally, many companies already offer APC solutions [13], yet their coverage area is still quite limited in the industry due to two main reasons: one of them has already been mentioned and is related to the widespread use of the PID controllers, and the other one is that the configuration and deployment of an APC solution is complicated and costly. This causes industrial stakeholders to only choose APC in cases where the application of the PID controller is not feasible at all. In this case, the return on investment and the desire to be able to automate a certain process loop can cover the APC implementation costs.

At the same time, the high costs of APC implementation leave out smaller players in the industry who cannot afford it. Furthermore, even with major players, smaller process control loops may also fall outside of the APC implementation scope, whereas Sustainable Development Goals (SDGs) stated by the UN [14] cannot be accomplished excluding smaller participants as their total amount is significant. Boiler houses, small- and medium-sized production facilities, data centers, commercial buildings, households, and a vast amount of other energy producers and consumers experience unnecessary losses due to inefficient energy usage. Leaving them out contradicts with SDG7.3 that assumes to “double the global rate of improvement in energy efficiency”. Meanwhile, buildings consume up to 40% of total energy produced, more than industry or transport [15,16].

Furthermore, another key performance indicator nowadays is the carbon footprint. The reduction of carbon emissions is especially important in major contributing industrial entities including energy generation processes and chemical production [17]. Model predictive control has been shown to be very effective in reducing carbon emissions in the industry [18]. On the other hand, it is also crucial to optimize the electricity and heating or cooling consumption in buildings. Towards that end, in [19], a 400% CO_2_ saving was reported for a single building following the introduction of a model predictive control strategy which optimizes the electrical and thermal loads.

Therefore, accelerating the rate of adoption of APC—and more precisely MPC—in various industries is seen as a rather pressing issue toward achieving the above-stated goals. As previously mentioned, one of the key elements of accelerating this rate is the introduction of efficient KTT mechanisms [20].

Academic investigations of optimization are frequently limited to simulations only, and the actual deployment of the solutions rarely takes place, resulting in rather low technology readiness of the whole solution [21]. University–Industry Technology Transfer (UITT) is a complex process, as bringing theoretical results to the industry involves risks that might be unacceptable to many industrial partners. One of the most significant hurdles to technology innovation is the high cost of innovation implementation and the relatively low level of R&D expenditure in commercial firms. This problem was outlined many decades ago when UITT appeared as a separate research field [22,23] and UITT still is a challenging procedure that requires special efforts and administrative support [24,25,26]. On the other hand, several highly successful examples of UITT can also be found in the literature [27,28].

A scheme of APC solution development and transfer from academia to industry is presented in Figure 1. The data collected from industry are used to develop control and optimization algorithms (research phase) in academia using the academic tools reviewed in this paper. The proposed MPC application is used as an example to solve the problem of transferring the developed algorithm in software from academia to industry and its implementation in a production environment [29].

We now state the contribution and novelty of the work presented in this manuscript. First, the results reported in [30] are extended by developing the general software framework for deploying APC algorithms to production environments. This is the most significant innovation that underpins the technology transfer aspect since it enables for the implementation of any type of controller while maintaining a consistent and smooth user experience with the same user interface. The second part of the contribution is concerned with the study of the implementation process of the MPC algorithm specifically and shows that the novel framework is well suited for facilitating technology transfer from academia to industry. The solution is deployed to an industrial plant the proper operation of which is critical since it serves a city district. The end result is a more stable control of an industrial process output. In fact, the application of the MPC improved the performance of the industrial control loop almost threefold compared to the original PID-based control according to several performance metrics. Other achieved benefits are the reduced number of manual interactions between operators and process control, as well as lower fuel use. Furthermore, as a result of process control optimization, the CO_2_ emission is also reduced, thus supporting the sustainable development goals.

The rest of the manuscript is structured as follows. The theoretical and practical aspects of developing and deploying the MPC algorithm are described in Section 2. The essentials concerning the development of the proposed solution are provided in Section 3. Section 4 describes practical aspects of APC technology transfer. Section 5 is dedicated to the application to industrial implementation. Section 5 is dedicated to the implementation of the developed solution to an industrial plant. Section 6 concludes paper with results and future development plans.

## 2. Model Predictive Control: Theoretical and Practical Aspects

In this work, model predictive control—a well studied and powerful control method—is considered as the advanced control approach. It is, however, also well known that it is far more difficult to transfer it to a production environment compared to, e.g., a PID controller. In the following section, the MPC method is briefly recalled.

### 2.1. MPC Theory

In MPC, the main aim is to minimize a cost function defined as [31]:(1)$$V\left(k\right)={∥Z\left(k\right)-T\left(k\right)∥}_{Q}^{2}+{∥\Delta U\left(k\right)∥}_{R}^{2},$$
where Z(k) is the outputs’ prediction vector within prediction horizon $H_{p}$, T(k) is the set points’ trajectory within $H_{p}$, and ΔU(k) is the vector of process input moves (changes) within the control horizon $H_{c}$.

After simple algebraic manipulation, the cost function can be rewritten as:(2)$$V\left(k\right)=\mathrm{const}-\Delta U{\left(k\right)}^{T}G+\Delta U{\left(k\right)}^{T}H\Delta U\left(k\right),$$
where ΔU(k) is the only unknown that should be found to minimize V(k). The expressions for *G* and *H* are known and are obtained from (Equation 1). The constant term const is eventually of no importance as it disappears during the differentiation of V(k) as part of the minimization procedure.

In case of MPC, it is possible to set constraints on input moves Δu, input value *u*, and output *z* as $\Delta u_{\min}\leq \Delta u\left(k\right)\leq \Delta u_{\max}$, $u_{\min}\leq u\left(k\right)\leq u_{\max}$ and $z_{\min}\leq z\left(k\right)\leq z_{\max}$ at any moment, where min and max are the lower and upper bounds of the corresponding variable. As the controller produces optimal input moves, then we are interested to express all equations through Δu.

The combined inequality has the form:(3)$$\left[\begin{array}{c} \Lambda \\ \Phi \\ \Gamma \end{array}\right]\Delta U\left(k\right)\leq \left[\begin{array}{c} \lambda \\ \phi \left(u(k-1)\right)\\ \gamma \left(x(k),u(k-1)\right) \end{array}\right],$$
where Λ, Φ, Γ are known polynomials and λ, ϕ, γ are known functions, u(k−1) is the process input vector at the previous time sample, x(k) is the model state vector at the current time sample, and ΔU(k) is the vector of future control moves known at the time moment *k*, see [31] for details.

Finally, taking into account (Equation 2), we get a problem in the quadratic programming form:(4)$$\min_{\theta }\frac{1}{2}\theta^{T}H\theta +h\theta$$
subject to Ωθ≤ω.

This type of problems can be solved using standard methods. Most popular are the active set and interior point method. In this work, the interior point method was used as it was developed later then the active set method and is known to require less computational power.

### 2.2. Interior Point Method for MPC

The interior point or barrier function method is used in the present MPC formulation. It is deeply studied in [31,32,33,34,35,36,37,38] and a brief overview is given next.

The barrier function method brings the problem in Equation (Equation 2)—excluding the constant component—to
(5)$$V\left(k\right)=\Delta U{\left(k\right)}^{T}H\Delta U\left(k\right)-\Delta U{\left(k\right)}^{T}G$$
subject to constraints (Equation 3) to the form with no explicit constraints. The modified cost function is given by:(6)$$V\left(k\right)=\Delta U{\left(k\right)}^{T}H\Delta U\left(k\right)-\Delta U{\left(k\right)}^{T}G+\mu B\left(\Delta U\left(k\right)\right),$$
where
(7)$$B\left(\Delta U\left(k\right)\right)=-\sum_{i}^{H_{u}m}\left(\ln\left(\Delta U_{\max_{i}}-\Delta U_{i}\left(k\right)\right)+\ln\left(\Delta U_{i}\left(k\right)-\Delta U_{\min_{i}}\right)\right).$$

In (Equation 7), *B* is a logarithmic barrier function, *i* refers to the *i*th element of vectors $\Delta U_{\max}$, $\Delta U_{\min}$, and ΔU(k). There are also other types of barrier functions, but the logarithmic function is the most convenient to use with standard optimization techniques. Function (Equation 6) is smooth, so it can be solved with a suitable formulation of the Newton’s method. As μ→0, the solution of (Equation 6) tends to the solution of (Equation 5), see [33].

With the application of Newton’s method, the update of ΔU results in
(8)$$\Delta U=\Delta U-\alpha H^{-1}g,$$
where *g* is the gradient vector of the cost function, H is the Hessian of the cost function and α is a scalar to ensure reduction of the cost function. Furthermore,
(9)$$g\left(i\right)=H\left(i,:\right)\Delta U\left(k\right)-G\left(i\right)+\mu \left(\frac{1}{\Delta U_{\max_{i}}-\Delta U_{i}\left(k\right)}-\frac{1}{\Delta U_{i}\left(k\right)-\Delta U_{\min_{i}}}\right),$$
where *i* refers to the *i*th element of gradient vector *g* and vector *G*, H(i,:) refers to the *i*th row of the matrix *H*. The Hessian matrix H is
(10)H=H+μD,
where *D* is a diagonal matrix with the following elements on the diagonal:(11)$$D\left(i,i\right)=\frac{1}{{\left(\Delta U_{\max_{i}}-\Delta U_{i}\left(k\right)\right)}^{2}}+\frac{1}{{\left(\Delta U_{i}\left(k\right)-\Delta U_{\min_{i}}\right)}^{2}}.$$

Gradient and Hessian expressions are only true if ΔU(k) is feasible. That requires the starting point for the optimization to be within the limits, which can be achieved, if, for example, only input constraints are used:(12)$$\Delta u\left(k\right)=\frac{u_{\max}+u_{\min}}{2}-u\left(k-1\right),$$
and
(13)$$\Delta u\left(k+i\right)=0,\;i=1,\ldots ,H_{u}-1.$$

In case of output constraints, a feasible starting point may not exist if these constraints contradict each other. In these circumstances, output constraints should be softened, meaning they should be shifted enough in the search space until a feasible point is reached. The reason for softening output restrictions is due to the physical limitations of input constraints, which makes it impossible to soften these restrictions [31].

For the implementation of the interior point method, one also needs a scaling factor ν∈(0,1) to prevent violations of the constraints, a positive integer *K* to define the number of iteration steps, a barrier scaling factor μ>0, and a barrier-scaling-factor weighting factor ζ. The closer the value of ν is to 1, the closer Δu(k) goes to the limit. The scaling factor μ is required to prevent a jump out of the feasible region at the beginning of the algorithm execution. After the initial value is assigned (e.g., to 10), it decreases in each iteration of the algorithm, thus leading to the convergence of the problems defined in (Equation 5) and (Equation 6). Note that the inversion of the Hessian matrix H is not needed as $\rho =H^{-1}g$ can be calculated with a QR-decomposition using the Householder algorithm [39].

If the optimal solution for the unconstrained MPC problem is located beyond the constraints, then in each iteration of the interior point method, one should decrease the value of μ and approach the value of the constraint without crossing it.

The algorithm that is ultimately used in the implementation of the MPC algorithm in the present work is depicted in Figure 2. First, the feasible initial values of the control moves are found using (Equation 12) and (Equation 13). Then, the iterative process to ensure the fulfillment of the constraints (*j*-steps) is started: the gradient vector *g* and Hessian matrix H of the cost function (Equation 5) are found and the next update of the control moves is calculated as ρ with Newton’s method. Next, all elements of ρ (*i*-steps) are tested for the control moves vector ΔU to be within the constraints. If any element of ρ violates the constraints, then its value is set to bring ΔU back into the region where the constraints are fulfilled. Meanwhile, the α parameter with α∈[0,1] is introduced to get a better estimation of the ΔU update. After this, the control moves ΔU and the barrier weight μ are updated, and the algorithm is repeated until *K* iterations are reached.

## 3. Application Development Tools and Structure

In this section, a summary is provided concerning the actual implementation of the above-mentioned MPC algorithm as part of a flexible intelligent control platform, for which the MPC approach is the preferred method, whereas the platform is capable of hosting any conceivable control method as well.

For building the application, Java programming language was selected because of its cross-platform support and wide adoption. Numerous IDEs (Integrated Development Environments) are available at no charge and provide sufficient convenience for the development of complex applications. We considered Eclipse [40] and IntelliJ [41] IDEs, but finally selected IntelliJ due to its greater coding assistance functionality.

A further advantage of using Java is the availability of a large number of freely available libraries that lets one focus on the task at hand without delving deeper into low level operations (such as communication protocols and methods, database access, etc.) that can be implemented from available, oftentimes unit-tested code. Java software dependencies were managed using the project management tool Maven [42]. Maven is another free software package that is maintained by Apache Software Foundation [43] to help manage and arrange the development of software components. The Maven configuration is done in the POM (project object model) XML file, which is easily filled with the proper settings with the help of IntelliJ’s project template set and autocomplete features.

The Spring Framework [44] was chosen as the basis for implementing the backend of the application. It provides mechanisms to build complex applications in a relatively simple manner [45]. Inversion of Control, Aspect Oriented Programming, Data Access framework, Context management are some examples of the features of Spring. It is also important that the framework is helpful for building and running modular software since the developed software platform is intended to be flexible and modular. Spring helps to avoid implementing a lot of low level operations, e.g., database access, as these operations are automated and require just a few lines of code to be implemented. Context management allows us to create independent software modules that are automatically combined on the fly into one application.

While Java was considered reasonable for the backend functionality of the application, we decided to use the Angular framework with Typescript for the user interface. The choice of creating a Web UI was dictated by modern day requirements and expectations, but Java frameworks for this type of solution are either premium or do not have adequate functionality. At the same time, Angular provides all of the necessary options to build modern web interfaces.

The main functions that the application has to cover include:Maintain a list of data points—measurements from the actual industrial process—the operator is going to be working with;Configure and run communication via Modbus/TCP protocol with the process control system;Configure and run the control algorithm;Save the configuration to the database whenever it is modified and load it from the database on start-up.

These functions are carried out in the backend of the application, while the frontend works from an internet browser and communicates with the backend to load or update the configuration. The structure of the application is shown in Figure 3. The web application can be considered a software client in relation to the backend. It uses the HTTP protocol to fetch values in JSON format from the backend and displays these in a suitable way in the operator’s web browser. All presented data can be edited and saved.

The backend contains an algorithm module where any algorithm can be implemented (in the case of this work, it is MPC). It takes input values from the data points list, makes computations, and saves output values as other data points in the same list. A communication thread runs separately. It reads input values from the process control system and saves them to configured data points, then reads output values from other data points and sends them to the control system.

The following subsections describe the application’s main functions in more detail.

### 3.1. Data Points

The backend allocates memory for a list of data points, which will be used as inputs and outputs for the algorithm. There are currently three types of data in the list: *double*, *integer*, and *boolean*. The data points list is displayed as a table in the web browser. Each row has Edit/Save and Remove buttons.

### 3.2. Communication

Communication is implemented as Modbus/TCP master using the JLibModbus Java library [46]. This library is available from Maven repositories; so, no extra import efforts are needed except configuring it in a Maven POM file. The choice of using Modbus stems from its relative popularity in industrial communication.

For Modbus master, we need to configure read and written data points of two types: coils (bit) and registers (words or 16 bit integers). It is possible to use two registers to code a 32 bit single precision real value (IEEE-754). It is sent as two registers and must be decoded on the other communication side as it is not a standard Modbus functionality.

Implementation of the user interface for protocol configuration is more involved. It consists of three levels. On the first level, we have a list of Modbus communication lines as we suppose possibility to use the application for communication with many systems simultaneously. Each line has an Edit button that opens the second level of the protocol line configuration parameters. This level includes also data point reads and writes each with its own Edit button that opens the third level with a list of Modbus addresses of each data item to be sent or received. Here, Modbus addresses are associated with data points list items. Addresses’ values read from the Modbus slave are saved to value fields of the configured data point. Values of written addresses are acquired from the value field of the configured data point and sent to the slave. A process control system has to be implemented as a Modbus slave in this configuration.

### 3.3. Web HMI

Independently of how the high level of automation is achieved in any production process, the final decision of the way to proceed is in the responsibility of a human. Any automation system has operators who follow the process and interact with corrective actions, if something goes wrong. The processes become bigger and more complex with time, so the amount of data to be processed becomes huge; hence, the main trends in building a modern human–machine interface (HMI) move toward efficient data presentation without overloading the operator with a stream of data that is very difficult to process or to understand [47,48].

In the case considered in this work, a limited amount of data is used in the application. We need to provide visualization of the communication protocol settings, required process parameter values, settings of the algorithm, and the outputs produced by the algorithm that are sent to the process control system. The Web HMI is used in the scope of this work as it has already become a standard HMI solution in many industrial use cases.

As was mentioned earlier, the Angular framework was used to develop the HMI—an open-source framework developed->supported by Google. Angular creates a stand-alone frontend that is compatible with any backend application supporting HTTP communication. It is possible to use any programming language for backend development including Java.

With Angular, one can choose from a variety of UI components that may display any type of information in the browser. For data sharing, Angular employs the widely recognized JSON format. As a result, the framework enabled the creation of a user interface for the MPC application that was both convenient and easily configurable. Parameters of the application are displayed as text. Many of them have matrix form which is easy to present as tables with suitable border formatting. For convenient modifications, matrices are converted into text boxes with values in Matlab format, e.g., [1 2 3; 4 5 6] for
(14)$$\left[\begin{array}{ccc} 1 & 2 & 3\\ 4 & 5 & 6 \end{array}\right].$$

The resulted User Interface example with MPC diagnostic page is presented in Figure 4.

## 4. Practical Aspects of the Implementation and Industrial Integration

Certain development steps must be performed for the MPC implementation to become an efficient practical solution at the final stage when it is deployed to control the industrial process [49,50]. Moreover, since the goal of the present contribution is to also provide technology transfer for the MPC control method, the implementation should be in line with a coherent TT model, see [20].

MPC implementation has five steps that relate to the general TT model as shown in Table 1. Accordingly adopted TT scheme is presented in Figure 5.

The process is thoroughly examined in the first step, and it is determined whether MPC is applicable in the selected situation. Often, it is sufficient to tune existing control loops to achieve satisfactory control performance. If this is not successful, however, then one needs to determine whether the performance bottleneck lies with the existing equipment. Manual process control by an experienced operator could be used in practice. If better performance is obtained with manual control, it is obvious that automatic control has growth potential. Unfortunately, empirically revealing potential opportunities is not always possible, hence a theoretical approach should also be used. In this case, we should keep in mind that MPC is more beneficial for processes with dead time, constraints, and multiple inputs and/or multiple outputs [50]. This step includes formulating the problem statement, discovering the state of the art, and the development of a candidate solution.

The next stage of general TT assumes validation in academia, which means in case of MPC implementation three basic steps: collection of experimental data from the plant under study; data-driven model development based on the collected data; and MPC design using the obtained model. Process data can often be collected relatively easily as many processes are equipped with *Historian databases*. Process inputs should be changed in plant testing to cause process outputs to fluctuate around the main operating point for the goal of collecting dynamic process data [51]. Model identification and MPC design do not present considerable challenges for a seasoned researcher in the academia since in the area of systems and control, there are several tools available for researchers, such as Matlab with relevant extensions including the System Identification toolbox and MPC toolbox. Also free identification solutions exist such as Python-based SIPPY [52]. However, challenges arise with the implementation of the real process since moving the controller from a Matlab simulation to the real process control system is not straightforward [53,54]. There exist Python-based MPC solutions, but these appeared recently and are in the beginning of their career mainly presenting laboratory test results [55,56].

The final step of the MPC implementation requires the integration of the developed solution into the control system of a real process, whether this means a programmable logic controller (PLC) or a distributed control system (DCS) implementation. Some DCS systems may already include MPC implementations [50], but this is always a commercial solution with a significant associated cost that is often not feasible for smaller process optimization. Migration from step four to step five with academic or open-source software can be difficult because no ready-made appropriate interfaces are available. The application framework proposed in this manuscript solves the problem of transferring the designed MPC into industrial implementation.

## 5. Industrial Implementation

### 5.1. Process

In this paper, we focus on the problem of transferring APC solutions from the research field to the control of an energy generation process in industry. Here, we consider a water boiler that is a part of a bigger combined heat and power (CHP) plant. The primary goal of the CHP plant is to produce heat for nearby cities.

Originally installed in 1978, the boiler produced 100 kcal/h (116.3 MW) of heat power. The specific model of the boiler is KVGM-100 [57]. Several major investments were made to renovate the boiler infrastructure during its lifetime; so, it is now equipped with modern measurement and control devices and is connected to DCS. All the control applications are implemented in the DCS on software level. Applications can be created using predefined or programmable function blocks and downloaded to the DCS without interrupting the process control. Still, control methods presently used in boiler control are quite conservative and are based on the PI control algorithm.

The output water temperature of the boiler is a controlled variable in the boiler’s main control loop. The fuel combustion process, the transfer of heat power from furnace to water and hot water flow to the boiler output are relatively slow due to the size of the equipment, so there is a significant delay between the flow of gas into the furnace (manipulated variable), and the measurement of the water temperature at the boiler output (controlled variable). Related process diagram is shown in Figure 6. Since the PI controller uses output errors for control, the delay affects its performance. Therefore, the integration time is set longer than the measurement delay to prevent permanent overshoot of the manipulated variable. Specifically, the integration time was set to 240 s. As the ideal PI control algorithm $c_{out}=K_{p}\left(e+\frac{1}{T_{i}}\int e\;dt\right)$ is implemented in the related function block, then the real integration time is almost 270 s ($K_{p}=0.9$). This causes the controller to react slowly to changes in set points and disturbances. Due to the above considerations, retuning the PID controller is not a feasible option. An opportunity to apply the MPC control algorithm arises naturally according to prior discussion.

### 5.2. MPC Implementation

According to the implementation procedure described in Section 4, a process investigation and preliminary MPC design was performed. From operators’ and automation engineers’ interviews, it was learned that the control loop under investigation is frequently set to manual mode and corrective actions by the operator are applied to get better performance. The reason for this is mostly the process dead time. Depending on boiler load, the delay from the system input to output can reach 4 to 6 min. This is a good prerequisite to assume control performance improvement with use of MPC.

The next step is to perform a plant test to collect time series data for model identification. As the process is not controlled in a stable manner, the input/output variations are sufficient for model identification without any special test signals. Historian data are collected to the database all the time. We acquired these data with a sampling time of 10 s. After analyzing the process, it was decided that a 1 min sample time is sufficient for model identification and later process control, as over 5 prediction steps cover the average 5 min process delay and mitigate its effect.

The process variables that mainly affect plant temperature output are: water inlet temperature, total water flow, and gas flow. So, the candidate model structure was selected to have three inputs and one output. Both total water flow and inlet temperature depend on the district heat network load and cannot be manipulated. Thus, two of three inputs are measured disturbances and only one is a manipulated variable—gas flow.

After the data were collected and preprocessed, several state-space models were identified in the Matlab Identification Toolbox using the prediction error minimization (PEM) identification method. These models were used to make initial MPC design in the Matlab MPC Toolbox. An MPC was created with a model of better fit while other models were used to simulate the process in the Matlab/Simulink environment in various scenarios. After achieving satisfactory results in Matlab, more realistic offline tests were performed with the acquired controller.

### 5.3. Offline Simulation

Offline simulation is required to test MPC functionality without affecting the real process [58]. This ensures that MPC computes adequate manipulated variable values to bring the controlled variables to the desired trajectories.

In our case, the first offline test was performed in a computer with a mathematical model of a process different from the model used in MPC. Real DCS software was used for process simulation. A state-space simulation model was implemented as a Java functional block. Simulation time was increased 15 times compared to real time to make testing faster and more convenient; thus, the simulation sampling time was 4 s instead of 1 min. MPC parameters were transferred from Matlab to our MPC application. Communication between DCS and MPC application was configured using the Modbus/TCP protocol. On this stage, MPC was pre-tuned in the same software that is used for the real process control.

After the MPC functionality was validated, the next offline test was performed on a real process without passing the controlled variables values to the process control. The functionality of the controller was observed during several hours. The controller showed adequate reaction to all process changes, so the process owner accepted the controller for the online implementation.

### 5.4. Online Implementation

The control performance of the plant has improved significantly following the implementation of the MPC control. Figure 7 shows a comparison of PID and MPC control performance. A temperature deviation of one degree is acceptable in plant output. The PID controller, however, fails to achieve this target, while MPC keeps the temperature close to the set point as required by the specifications. This can be easily seen in the time series shown on the left side of the figure.

Furthermore, the error resulting from PID control has a wider distribution than that of the error stemming from the use of MPC. Its variation is far beyond the allowed limits as can be seen from the violin plot on the right of Figure 7. The main parameters for evaluating the controller performance are shown in Table 2. The MPC-driven control loop has nearly three times lower mean square error and sum square error than the PID one. The median value of its error is also lower than that of PID, which is not a global characteristic, but an advantage in this experiment, since it results in lower gas consumption in case of MPC. This will be discussed later in more detail.

Furthermore, we have to discuss operating points of the process at which we compare PID and MPC loop performance. The points are not the same as there are certain variations in the set point due to changes in hot water demand and certain ambient conditions, mainly weather. It is not enough to compare performance directly, we need to discuss the comparison environment as well. For that reason, timeseries were selected from source data that demonstrate the performance of the control loop in operating areas that are as close to each other as possible. The operating area is determined by the process inputs and the set-point value selected for the control. The process inputs relationship is depicted in Figure 8, in which we can see that the selected operating areas intersect.

The numerical ranges of operating areas are presented in Table 3. As can be seen, the set point is different for compared cases, but according to the end user, the MPC performance is clearly better at every operating point, since the PID could never reach the same result with low output variation. It was not possible to test the PID controller and MPC under identical settings because the tests were conducted at different times of the year, when the operational environment also differed, but efforts were exhibited in the present work to achieve a fair comparison of the results.

A further improvement in control performance would be achieved by reducing actuator hysteresis, which would result in the actuator moving more frequently, resulting in faster valve wear and would require more frequent maintenance cycles. Thus, the acquired performance is a compromise with the mechanical condition of the control equipment. The implementation of the new control loop caused operators to stop interacting with it manually. Overall, the control performance was adequate to avoid quality penalties in the future. In the end, the industrial stakeholders were satisfied with the obtained results.

Finally, it is important to evaluate gas consumption in two scenarios. As indicated earlier, it cannot be compared directly as the PID controller and MPC were tested under slightly different conditions (different set points, different water flows); however, cumulative output error provides a general estimate of gas consumption. Figure 9a illustrates the cumulative error of the plant output in relation to the lower limit (−1) of acceptable error value. The graph demonstrates that the cumulative error of the plant output temperature is higher with PID control. However, from a physics perspective, MPC would provide a reduction in gas consumption since it avoids heating water to unnecessarily high temperatures.

After normalizing the PID gas consumption to the MPC operating area by dividing it with in/out temperature difference and water flow of the PID operating area and multiplying by the same parameters of the MPC operating area, we get the gas consumption comparison in Figure 9b. The consumption in case of the PID controller is 1977 Nm^3^ higher after 2000 min of operation (here, Nm^3^ means the quantity of Natural Gas which, when absolutely dry, at a temperature of 0 °C and at an absolute pressure of 101,325 bar, occupies the volume of one cubic meter).

## 6. Conclusions and Discussion

In the present manuscript, the complete process of technology transfer from development to production was presented for an advanced process control method—the model predictive control method. In the proposed application [59], the linear MPC algorithm is implemented, and complemented with a communication and HMI infrastructure, which constitutes a complete framework for various algorithms to be developed and deployed in the industry. The application can be extended in the future to handle nonlinear models and nonlinear control or implement arbitrary new algorithms. The complete planned application structure from [30] is presented in Figure 10 with the current contribution highlighted. Intelligent Control System CORE and Model Predictive Control functionality have been implemented in the scope of the current work, the other algorithms will become available in the future. The crucial point is that all of those advanced control algorithms will support a smooth technology transfer due to the overall architecture and implementation of the framework presented in this paper.

The technology transfer resulted in a significant improvement of control system performance in case of a combined heat and power plant. Namely, MPC was applied to a gas-fueled boiler. As a result, the fluctuation of the temperature of the outgoing water has been minimized which also led to the reduction of wasted energy and, as a consequence, to minimal CO${}_{2}$ emissions.

Some items for future research and development are outlined next. First, the communication protocol is Modbus/TCP master-only, which limits communication options. The Modbus/TCP slave functionality will be added, as well as OPC UA, which is becoming increasingly popular in industrial systems. An improvement that would increase application usability is embedded model identification. This would require significant effort for application development, integration of time series database (e.g., InfluxDB [60] or TimescaleDB [61]), and implementation of cutting-edge identification algorithms. This would allow the application to be self-sufficient in the context of the advanced control of real industrial processes. It will also provide the necessary components to implement other control algorithms as mentioned previously.

## Figures and Tables

**Figure 1 sensors-22-04149-f001:**
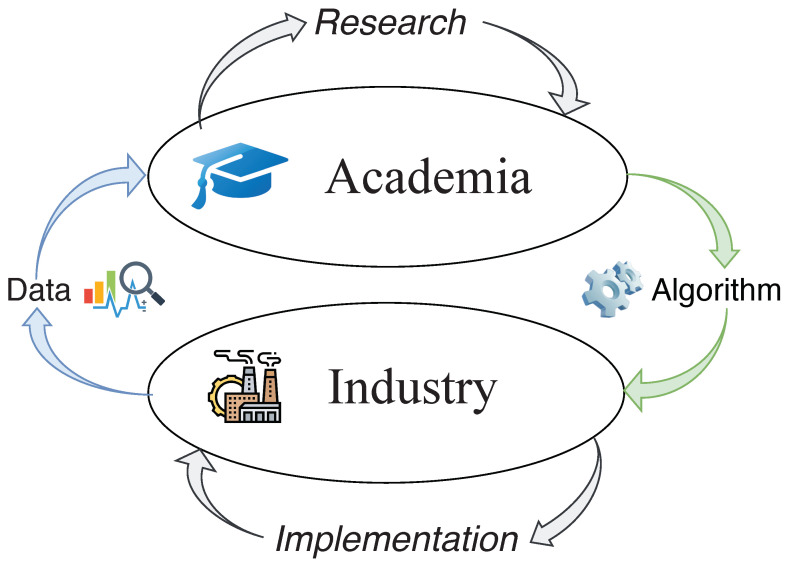
A typical cycle of advanced process control development: data are collected on the factory floor and shared with academic researchers; the data are processed, relevant research is conducted, and a control algorithm is designed and finally transferred back to the industrial plant where it is then implemented.

**Figure 2 sensors-22-04149-f002:**
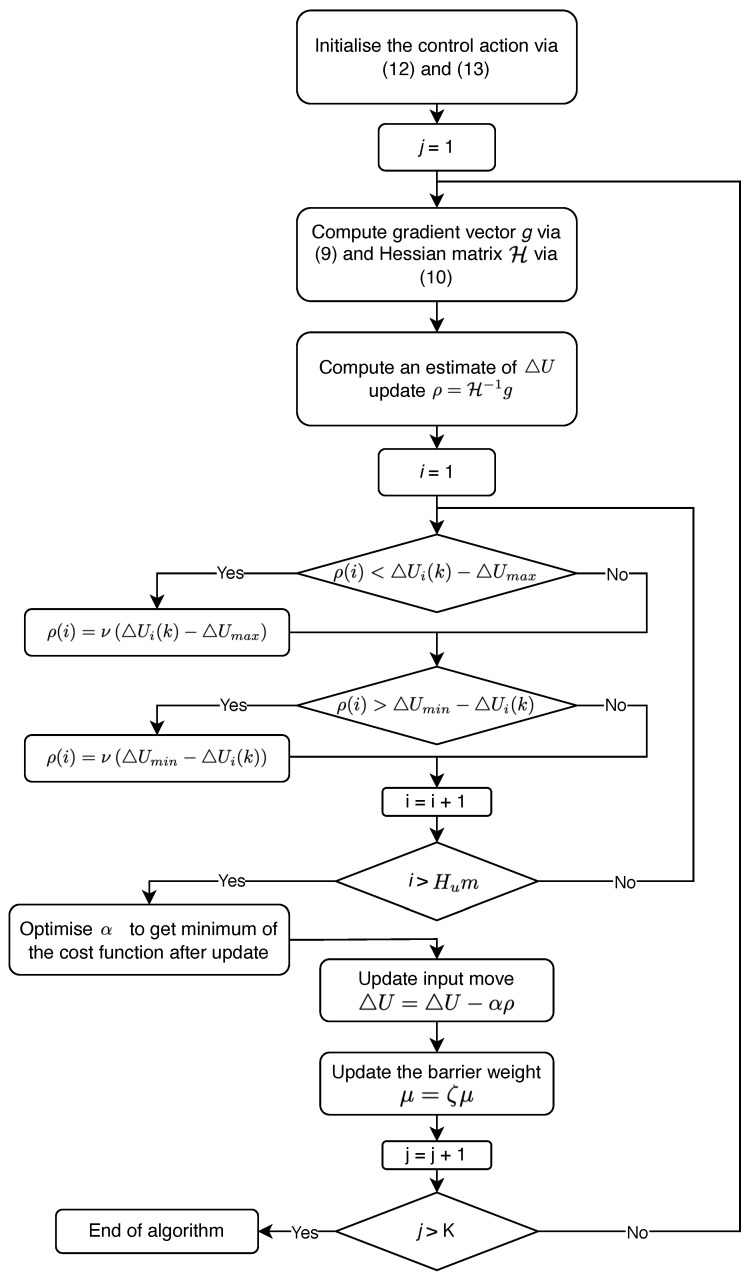
An algorithm for the implementation of constrained model predictive control.

**Figure 3 sensors-22-04149-f003:**
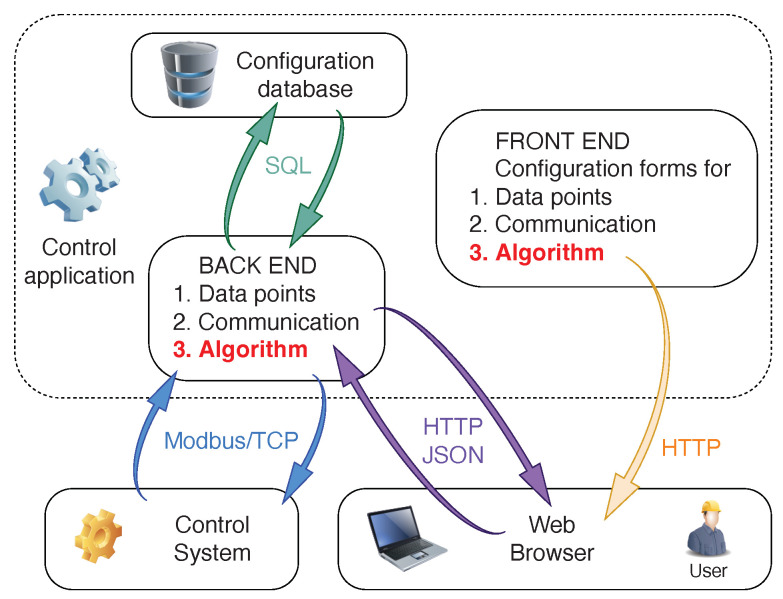
The structure of the developed advanced process control framework. The framework consists of the backend, which is responsible for communicating with a configuration database and the actual industrial control system, and the frontend, which is how users interact with the backend via an intuitive HMI running in the user’s browser.

**Figure 4 sensors-22-04149-f004:**
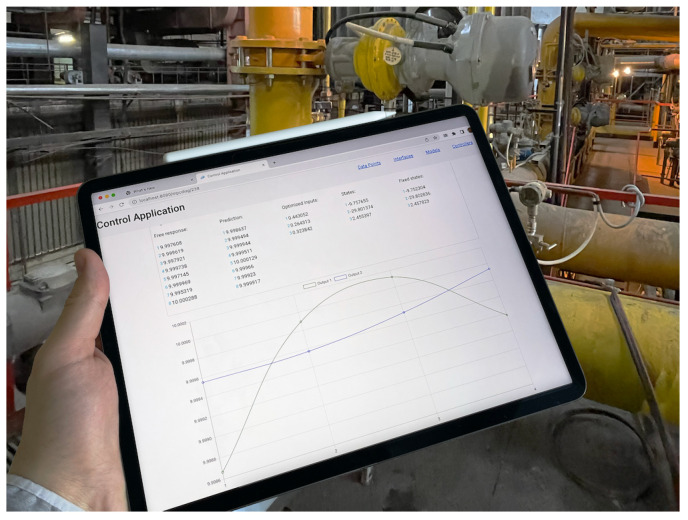
The User Interface of the developed MPC application as used on the factory floor.

**Figure 5 sensors-22-04149-f005:**
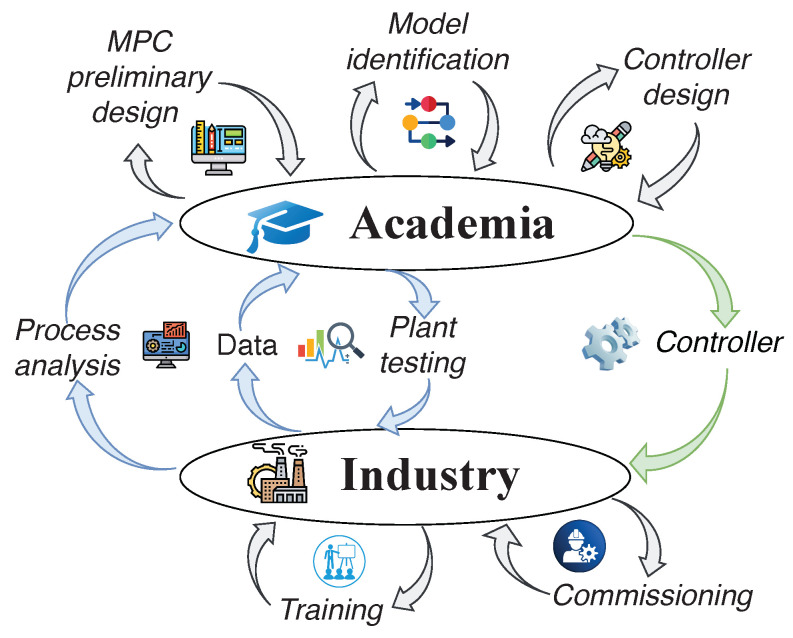
The process diagram of the proposed MPC transfer from academia to industry.

**Figure 6 sensors-22-04149-f006:**
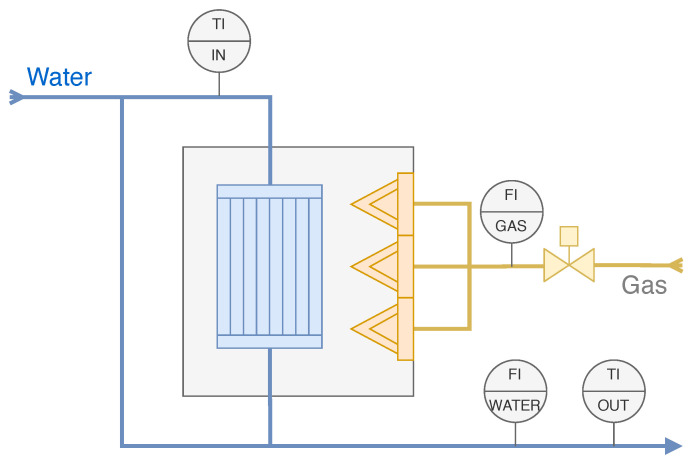
The process diagram of the considered industrial plant—the KVGM-100 boiler in the configuration for district heating whereby the passing water is heated with gas.

**Figure 7 sensors-22-04149-f007:**
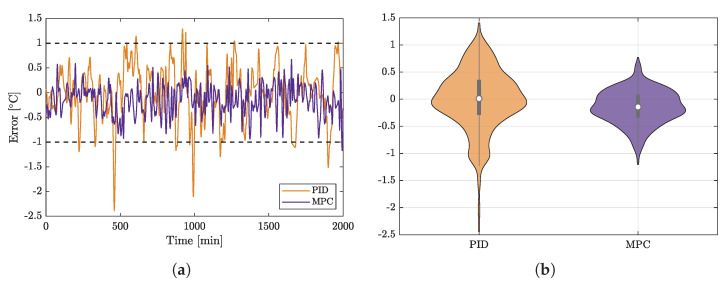
Performance comparison of PID and MPC-based control loops. (**a**) Timeseries showing the dynamics of set-point tracking error; the maximum tolerable deviation from the set point is ±1 °C, the closer the values are to zero, the better. The PID-based control loop frequently escapes the allowed error band. (**b**) Violin plots showing the distributions of deviations from the set point. It can be clearly seen that the MPC-based control loop results in a set point deviation that is more tightly packed around the origin, thus demonstrating the benefit of this control configuration compared to the PID control loop. (**a**) Timeseries showing the deviations from the set point for the compared control algorithms. (**b**) Violin plots for the set point deviations’ distributions for the compared control methods.

**Figure 8 sensors-22-04149-f008:**
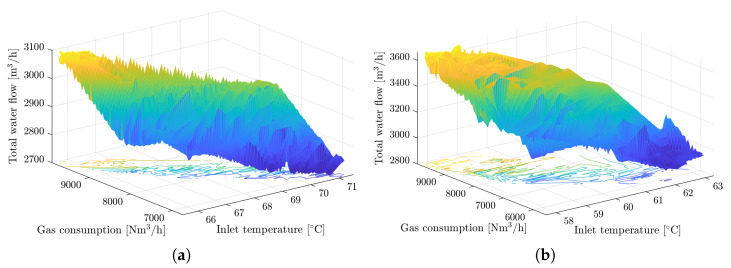
Surfaces depicting the operating areas of MPC (left) and PID (right) control loops. Each axis represents an input to the system: the total water flow, gas consumption, and inlet temperature. This visualization aims to demonstrate that the operating areas for the MPC and PID control loops are similar and thus yield a fair comparison of the performance of both control loops. (**a**) A surface depicting the operating area of the MPC control loop. (**b**) A surface depicting the operating area of the PID control loop.

**Figure 9 sensors-22-04149-f009:**
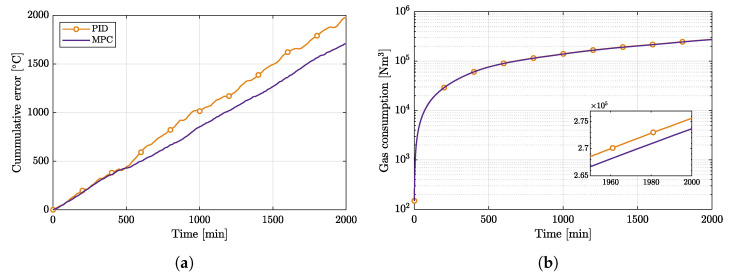
A comparison of gas consumption versus the cumulative control error for both MPC- and PID-based control loops. The gas consumption is reduced in case of the MPC controller; this becomes evident from the enlarged part of the right-hand plot. (**a**) Comparison of the cumulative control error (lower is better). (**b**) Comparison of gas consumption (lower is better).

**Figure 10 sensors-22-04149-f010:**
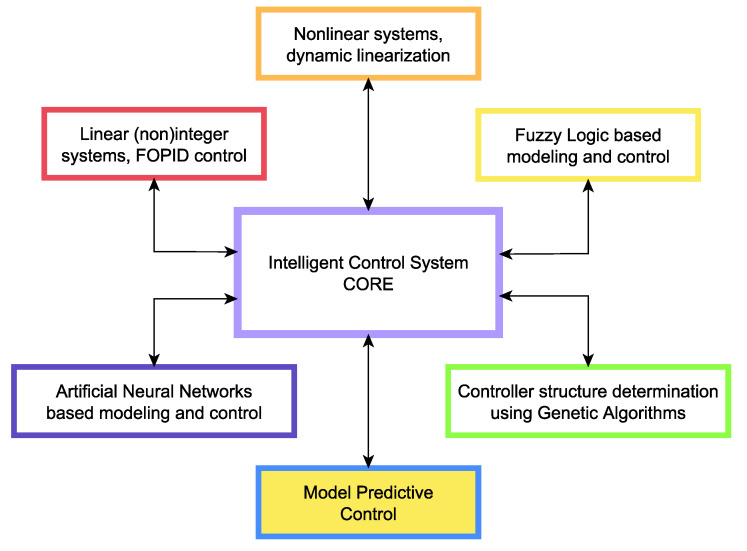
The overall framework developed in the scope of the present effort. The MPC module discussed in this paper is highlighted.

**Table 1 sensors-22-04149-t001:** The relation of the technology transfer model [20] to the MPC implementation steps as viewed from the perspective of designing and deploying process control.

Steps	General TT Model	MPC Implementation	Steps
1	Problem/issue	Process analysis and	
2	Study state of the art	preliminary MPC	1
3	Candidate solution	design	
		Plant testing	2
4	Validation in academia	Model identification	3
		Controller design	4
5	Static validation	Commissioning	
6	Dynamic validation	and	5
7	Release solution	training	

**Table 2 sensors-22-04149-t002:** Quantitative performance comparison of the PID- and MPC-based control loops based on set-point deviation.

	Mean Square	Sum of Squares	Median
PID	0.3173	634.5972	0.0114
MPC	0.1199	239.7205	−0.1421

**Table 3 sensors-22-04149-t003:** A summary of the operating areas for the MPC- and PID-based control loops. The operating areas depend on the set point, and three inputs to the system. The control loops are compared according to the operating areas specified in the table.

	Inlet Temp., °C	Water Flow, m^3^/h	Gas Flow, Nm^3^/h	Output Temp. SP, °C
PID	58–64	3000–3600	6000–10,000	78
MPC	66–72	2800–3200	7000–10,000	90

## Data Availability

Not applicable.

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
