# Peer review of "Bridging the Gap in Technology Transfer for Advanced Process Control with Industrial Applications"

_sensors, 2022, doi:10.3390/s22114149_

Round 1
Reviewer 1 Report
The authors analyze the link between research and industry in terms of applicability of advanced process control (APC), specifically model predictive control (MPC) evaluating the technology transferability.
The paper is sufficiently well written even if there are some sentences that need further explanation.
The overall structure of the paper is solid, but there are parts that can be shortened/expanded.
A list of the main comments/doubts/request is as follows:
1. Introduction
This is should be structured in terms of what is the goal of the paper and in this sense, this section lacks some narrativity to help the reader fully appreciate the effort of the authors. In addition:
- When speaking of standard control loops and Industry 4.0, it is also important to underline recent applications about process monitoring that can be achieved with appropriate strategies of data collection and cloud storage system (see Bacci di Capaci R. and Scali C. 2020. A cloud-based monitoring system for performance assessment of industrial plants. Industrial & Engineering Chemistry Research, 59(6), pp.2341-2352.)
- Another important family of techniques that is part of the APC is real-time optimization (RTO). The authors should mention it since this methodology has been used for more than 30 ys both in both industry and academia and used different optimizations strategies to calculate optimal economic setpoints that are often tracked by MPC (i.e. Cutler, C.R. and Perry, R.T., 1983. Real time optimization with multivariable control is required to maximize profits. Computers & chemical engineering, 7(5), pp.663-667.)
In this sense a recent application of RTO to a chemical facility using an ad-hoc connections system via TCP has been performed in Vaccari, Marco, et al. "Optimally Managing Chemical Plant Operations: An Example Oriented by Industry 4.0 Paradigms." Industrial & Engineering Chemistry Research 60.21 (2021): 7853-7867.; Badii, Claudio, et al. "Industry 4.0 synoptics controlled by iot applications in node-red." 2020 International Conferences on Internet of Things (iThings) and IEEE Green Computing and Communications (GreenCom) and IEEE Cyber, Physical and Social Computing (CPSCom) and IEEE Smart Data (SmartData) and IEEE Congress on Cybermatics (Cybermatics). IEEE, 2020.
2. Model Predictive Control: Theoretical and Practical Aspects
- Eq (2): what is const? Please expand it to make the reader understand better. The minimum is to writhe const(\DeltaU(k)) in which const(\DeltaU(k) is defined as…
- lines 122-127: why is needed such evident explanations on unconstrained MPC? One of the major advantages of using MPC is constrained optimization, hence I would remove such obvious lines.
- line 130: x_min, x_max are not fully explained and can be misleading. Please rephrase it using another letter and indicating it as a general variable or simply defining \min and \max as “the lower and upper bounds of the corresponding variable”
- Eq (3): x(k) is not defined. The authors should define the MPC problem once and for all at the beginning of this section so they can then can develop the modifications they need to underline. For example min_x,u (V) s.t. y = h(x), x+= g(x,u)
- when referring to Interior point methods is important to consider also the IPOPT algorithm (Wächter, Andreas, and Lorenz T. Biegler. "On the implementation of an interior-point filter line-search algorithm for large-scale nonlinear programming." Mathematical programming 106.1 (2006): 25-57.)
3. Application Development Tools and Structure
- Fig. 2.: typo: adnvaced
4. Practical Aspects of the Implementation and Industrial Integration
- lines 262-263: I do not understand this sentence, what are the authors referring to with “development process”? please rewrite and clarify
- lines 287-288: When referring to model identification toolboxes the author should also consider for reference a recent open-source toolbox that has been applied in different industrial fields (energy management, pulp and paper, chemical processes): see SIPPY (https://github.com/CPCLAB-UNIPI/SIPPY/wiki)
5. Industrial implementation
- line 309: DCS has been already defined
- lines 348-350: please explain what you mean by “One model was used for process simulation and other models for MPC design”, and explain what kind of models have been identified and with which SS method.
- Table 3. Why PID and MPC are tested on different set points 78°C and 90°C? Are those set points already achievable by both controllers based on the surfaces depicted in Fig 6?
Author Response
Please find attached the file with detailed answers to the raised issues.

Reviewer 2 Report
I have the following observations:
- In this work it is approaches the Gap in Technology Transfer for Advanced Process
Control with Industrial Applications and a potential bridging for resolving.
- Please insert a Paragraph, where to present more clearly for the readers, (preferably in Introduction) yours contributions in this paper
- Please insert in References a more papers from last 5 years.
- Please summarize the superiority yours algorithms in comparison with others, through imposing a few comparative quantities.
- The article if properly developed would be more suitable for “review” than “article”. In any case, the approach is very general. Indeed, when describing software, it is difficult to find a suitable wording for an article, because there is a large amount of information to be presented, and the presentation turns into a description of a project. For the presentation of algorithms, however, it should be refined so as to eliminate ambiguities from the general presentation (for example "After simple algebraic manipulation the cost function can be rewritten as: (2)" but who's G ?, etc.).

Author Response

(The authors gave the same response as above.)

Reviewer 3 Report
Introduction:
- presents the general vision of the Industry 4.0 and the connection with the optimization and the transition to the efficient modes of industrial operation;
- I agree with the idea that the the majority of existing control systems use PID conventional controllers;
- also, the considerations about the APC are to be taken into account;
- then, the authors refers to the technology transfer from the university to the industry as a complex process, with a suggestive diagram;
- according to the authors, the novelty of the paper refers to: developing the general software framework for deploying APC algorithms to production environments; the implementation of the MPC algorithm, used for the technology transfer from university to industry; the investigation of the solutions in a combined heat and power plant.
2. Model Predictive Control: Theoretical and Practical Aspects
- presents: MPC Theory and Interior Point Method, including the cost function, constrains etc.
3. Application Development Tools and Structure
- presents the actual implementation of the MPC algorithm as part of an intelligent control platform (using Java) and the main functions that
the application have to cover;
- the structure of the application (Fig. 2) highlights the backend, the frontend and the connection to the control system;
- in this context, the User Interface of the MPC application is a must;
4. Practical Aspects of the Implementation and Industrial Integration
- the steps of the technology transfer are very well described in the section and presented in Table 1;
- here, according to the authors, the migration from step four to step five with academic or open-source software can be difficult.
5. Industrial implementation
- the process used for test is a combined heat and power (CHP) plant;
- the necessary data about CHP are presented in this section, including the behaviour with a PI control law and the problems caused by the dead time;
- the identification of the process based on inputs/outputs are a proper one;
- in MPC versus PID control, the advantages of the MPC are obvioses in keeping the temperature close to SP;
- a very fair approach is to discuss about the comparison environment (Fig. 6);
- the obtained results (Table 3) and, very important, the fact that the industrial stakeholders were satisfied with the obtained results are a reward for the authors' effort.
Finally, the Conclusions presents the main contributions of the paper and the future developments.
Author Response

(The authors gave the same response as above.)

Round 2
Reviewer 1 Report
I appreciate the effort that the authors showed in addressing most of my comments, nevertheless, some additional changes/clarifications can improve the paper quality:
- The references suggested and added in the Introduction should be removed as clustered and expanded to help the narrativity as done for ref [6] of the revised paper version
- Please remove the separation in subsections 1.1 and 1.2
- A clear explanation of the MPC algorithm adopted should be stated in section 2.1. The algorithm in Figure 2 can help in the description of the equations.
- I appreciated the introduction of Figure 2 but this it is not yet referenced in the paper anywhere as for Figures 4, 5 and 6. Every figures or tables should be referenced in the text ohterwise it should be removed
Author Response
Please find attached the response letter with detailed replies.
